# A New Strategy for the Old Challenge of Thalidomide: Systems Biology Prioritization of Potential Immunomodulatory Drug (IMiD)-Targeted Transcription Factors

**DOI:** 10.3390/ijms241411515

**Published:** 2023-07-15

**Authors:** Thayne Woycinck Kowalski, Mariléa Furtado Feira, Vinícius Oliveira Lord, Julia do Amaral Gomes, Giovanna Câmara Giudicelli, Lucas Rosa Fraga, Maria Teresa Vieira Sanseverino, Mariana Recamonde-Mendoza, Lavinia Schuler-Faccini, Fernanda Sales Luiz Vianna

**Affiliations:** 1Graduate Program in Genetics and Molecular Biology, Genetics Department, Universidade Federal do Rio Grande do Sul (UFRGS), Porto Alegre 91501-970, Brazil; mfeira@hcpa.edu.br (M.F.F.); ggiudicelli@hcpa.edu.br (G.C.G.); msanseverino@hcpa.edu.br (M.T.V.S.); lschuler@hcpa.edu.br (L.S.-F.); 2Teratogen Information System (SIAT), Medical Genetics Service, Hospital de Clínicas de Porto Alegre (HCPA), Porto Alegre 90035-903, Brazil; lrfraga@ufrgs.br; 3Laboratory of Genomic Medicine, Center of Experimental Research, Hospital de Clínicas de Porto Alegre (HCPA), Porto Alegre 90035-903, Brazil; vlord@hcpa.edu.br (V.O.L.); juliagomes@hcpa.edu.br (J.d.A.G.); 4Bioinformatics Core, Hospital de Clínicas de Porto Alegre (HCPA), Porto Alegre 90035-903, Brazil; mrmendoza@inf.ufrgs.br; 5Biomedical Sciences Course, Centro Universitário CESUCA, Cachoeirinha 94935-630, Brazil; 6Post-Graduation Program in Medicine, Medical Sciences, Universidade Federal do Rio Grande do Sul (UFRGS), Porto Alegre 90035-003, Brazil; 7Department of Morphological Sciences, Institute of Health Sciences, Universidade Federal do Rio Grande do Sul (UFRGS), Porto Alegre 90010-150, Brazil; 8School of Medicine, Pontifícia Universidade Católica do Rio Grande do Sul (PUCRS), Porto Alegre 90619-900, Brazil; 9Post-Graduation Program in Computer Science, Institute of Informatics, Universidade Federal do Rio Grande do Sul (UFRGS), Porto Alegre 91501-970, Brazil

**Keywords:** lenalidomide, pomalidomide, cereblon, bioinformatics, zinc finger, microarray, RNA-seq, teratogen, transcriptome, graph

## Abstract

Several molecular mechanisms of thalidomide embryopathy (TE) have been investigated, from anti-angiogenesis to oxidative stress to cereblon binding. Recently, it was discovered that thalidomide and its analogs, named immunomodulatory drugs (IMiDs), induced the degradation of C2H2 transcription factors (TFs). This mechanism might impact the strict transcriptional regulation of the developing embryo. Hence, this study aims to evaluate the TFs altered by IMiDs, prioritizing the ones associated with embryogenesis through transcriptome and systems biology-allied analyses. This study comprises only the experimental data accessed through bioinformatics databases. First, proteins and genes reported in the literature as altered/affected by the IMiDs were annotated. A protein systems biology network was evaluated. TFs beta-catenin (CTNNB1) and SP1 play more central roles: beta-catenin is an essential protein in the network, while SP1 is a putative C2H2 candidate for IMiD-induced degradation. Separately, the differential expressions of the annotated genes were analyzed through 23 publicly available transcriptomes, presenting 8624 differentially expressed genes (2947 in two or more datasets). Seventeen C2H2 TFs were identified as related to embryonic development but not studied for IMiD exposure; these TFs are potential IMiDs degradation neosubstrates. This is the first study to suggest an integration of IMiD molecular mechanisms through C2H2 TF degradation.

## 1. Introduction

The lack of proper research on and the understanding of reproductive toxicology had severe consequences in the 1960s. Marketed as a safe drug, thalidomide sales around the world were used as a panacea [1]; however, its use in early pregnancy occurred soon after its release associated with the outburst of babies born with a range of congenital anomalies [2], later named thalidomide embryopathy (TE). It is believed that TE affected ten-thousand children worldwide before it was withdrawn from the market in 1962 [1]. TE is especially characterized by limb anomalies; however, the drug can compromise the correct development of almost every organ and system [3,4]. Multidisciplinary research has focused on the attempt to encounter a safe alternative for thalidomide [5]. The need is urgent once thalidomide immunomodulatory and anti-angiogenic properties led to its approval for multiple-myeloma treatment worldwide [6], and the other immunomodulatory drugs (IMiDs) synthesized later, lenalidomide and pomalidomide, are also used for treating this condition and are teratogenic in animal models [7,8,9]. In Brazil, thalidomide has been mainly used for erythema nodosum for leprosy (ENL) treatment since 1965 [10,11,12]. Although the studies demonstrate most individuals in treatment with thalidomide for ENL are male [13,14], a percentage of the subjects refer to women of a reproductive age [13].

Past researchers have discovered relevant molecular mechanisms that might be involved in TE, including anti-angiogenesis [15,16], increased apoptosis through the induction of oxidative stress [17], and binding to the cereblon protein, which is part of an E3-ubiquitin–ligase complex, named CRL4-CRBN [18]. The latter has been the main focus of the research lately because of the identification that CRL4-CRBN induces the degradation of Spalt-like transcription factor 4 (SALL4) when in the presence of thalidomide [19,20]. SALL4 is a zinc-finger transcription factor (TF) containing a C2H2 domain, and it is known for its role in limb development [19,20]. Since the discovery of thalidomide-induced SALL4 degradation, several researchers have evaluated other C2H2 TFs, which might also be neosubstrates of IMiD-induced degradation, through the CRL4-CRBN complex [21,22]. Neosubstrates is a term that has recently been applied to the IMiD research field to denominate CRL4-CRBN targets that are only degraded in the presence of one of the IMiDs [22], meaning these proteins are not physiological substrates of the complex. However, degrome screening performed by Sievers et al. (2018) only identified 11 zinc-finger neosubstrates [23], despite having more than seven hundred C2H2 zinc-finger TFs registered [24]. Moreover, other studies have focused on non-zinc-finger neosubstrates, such as tumor protein 63 (p63) [21] and heart and neural crest derivatives expressed 2 (HAND2) [25], or even non-TF proteins, such as casein kinase 1 alpha 1 (CK1α) [26]. Hence, this study aims to prioritize TFs that might be neosubstrates of the IMiDs thalidomide, lenalidomide, and pomalidomide, through bioinformatic combined strategies. Systems biology and differential gene expression analyses were performed after rigorous, systematized literature and database research, only comprising the experimental data. The results demonstrate beta-catenin (CTNNB1) as a hub in the network of thalidomide-affected proteins and specificity protein 1 (SP1) as a feasible C2H2 thalidomide neosubstrate. New C2H2 potential neosubstrates were also identified, resulting from an IMiD widespread effect on gene expression.

## 2. Results

A scheme of the strategy developed in the present study is described in Figure 1.

### 2.1. Literature Reports of 221 Genes and 80 Proteins Impacted by IMiD Exposure in Embryonic Cells/Tissues Were Identified

The literature review provided 407 entries from 48 manuscripts regarding the genes and proteins affected by the IMiDs during embryonic development (Appendix A). Most of these entries (n = 339) were exposed to thalidomide, followed by pomalidomide (n = 42) and lenalidomide (n = 26). Impacts on genes were reported in 253 of the 407 entries (62.1%), and 133 of these entries were based on human embryonic stem cell (hESC) assays. The remaining entries (n = 154) referred to the impact on proteins, reported especially in human umbilical vein endothelial cells (HUVECs; n = 51). Across the studies, 23 genes and 34 proteins were replicated, meaning these genes/proteins were detected as altered by the IMiDs in two or more studies. Hence, excluding those replicates, there were 221 distinct genes and 80 distinct proteins impacted by IMiD exposure, according to the literature review; only 20 targets were reported to be affected both at the gene and protein levels. The most annotated gene in the literature review was vascular endothelial growth factor A (*VEGFA*) (n = 5), whilst the most reported protein was cereblon (n = 11). The VEGFA protein was also cited five times as being altered by IMiD exposure; however, no reports regarding the effects on the *CRBN* gene were encountered. Impacts on genes were mostly related to the expression profile, with 174 reports of downregulation and 78 reports of upregulation. Protein impacts were also related to downregulation (n = 52) and protein degradation (n = 38). The literature review demonstrated evidence that IMiDs’ impacts on embryonic cells or tissues were more related to alterations in gene expressions, with *VEGFA* being the most studied gene. However, the reports showed that the studied IMiD effects on proteins were also considerably based on studying the neosubstrates’ degrome mechanism, especially driven by the amount of research on the CRBN protein.

### 2.2. Beta-Catenin Is an Essential Protein in the Network of IMiD-Affected Proteins

The 80 proteins reported in the literature review (Appendix A) to be affected by the IMiDs were inserted in the STRING tool to obtain a protein–protein interaction network, then transferred to Cytoscape v.3.7.2 software for topological analysis. Forty-six proteins (nodes) presented at least one interaction (edge), and 36 of these proteins were arranged in a main network with 51 edges, a clustering coefficient of 0.334, and an average number of neighbors of 2.833 (Appendix A). The full statistics of the main network are presented in Appendix A. The ten proteins that did not interact with the nodes of the principal network were excluded from the subsequent analyses.

After the network topological analysis, beta-catenin (CTNNB1), a non-C2H2 TF, was considered the essential protein of the network, being first ranked in degree, closeness, betweenness, and maximal clique centrality (MCC) (Table 1); the complete results, for all the nodes, are presented in Appendix A. This result indicates that CTNNB1 is (I) a hub based on the high degree; (II) a node located where most of the information flows in the network, being a possible controller of this information, based on the high betweenness centrality; (III) a node through which the information flows faster, based on the high closeness centrality; and (IV) an essential protein to the network, based on the centralities’ measures, including the maximal clique centrality (MCC). Other high-ranked proteins were CRBN and CUL4A, both members of the CRL4-CRBN complex.

Figure 2 presents the main network obtained. The sizes of the nodes are based on the MCC score and the colors of the nodes show the molecular mechanisms these proteins are related to; these mechanisms were obtained through the literature review. When analyzing the network configuration, it can be concluded that the CTNNB1 central position might contribute to the propagation of a systemic effect induced by the IMiDs. Hence, thalidomide binding to the CRL4-CRBN complex (represented by the yellow color) might induce alterations in molecular signaling pathways. From this perspective, through CTNNB1, these signaling alterations can be reflected in the angiogenesis mechanisms (orange) or cell cycle (blue). There is no evidence implicating that CTNNB1 is a neosubstrate of the CRL4-CRBN complex. However, its central position in the network suggests that molecular alterations induced by the IMiDs might be easily reflected in beta-catenin and, through this protein, be transmitted to other non-neosubstrate proteins. This mechanism helps to explain the IMiDs’ systemic effects on embryonic development.

### 2.3. IMiDs’ Exposure Results in a Widespread Impact on Gene Expression

IMiD-induced effects on gene expression were studied using a literature review and by the analysis of transcriptomes available from the GEO repository. As presented before, the literature review provided 221 genes affected by the IMiDs, with *VEGFA* being the most annotated gene (Appendix A). The search performed in GEO provided 166 datasets. Forty-five datasets met the eligibility for analysis: 11 with thalidomide exposure, 29 with lenalidomide exposure, and five with pomalidomide exposure. Twenty-two studies were further excluded because of the low quality of the samples, or because no significant differential gene expression (DGE) was detected (Appendix A). The differentially expressed genes of the 23 remaining datasets were annotated, providing 8624 differentially expressed genes across the studies (Appendix A). This number represents 30.4% of the genes contained in the reference genome (GRCh38/hg38) used for alignment (28,395 genes) (Table 2). The coagulation factor XIII A chain (*F13A1*) was the gene most frequently differentially expressed, significantly altered in nine datasets. Galectin-binding protein 3 (*LGALS3BP*), implicated in immune response, and Serpin family H member 1 (*SERPINH1*), a collagen-specific chaperone, were differentially expressed eight times. Of the 221 genes reported in the literature, 102 (46.1%) were differentially expressed in at least one dataset that was analyzed (Appendix A). The DGE of the most reported literature gene, *VEGFA*, was found in three datasets that we analyzed. Following the literature, the *CRBN* gene did not present significant expression alterations in any of the datasets analyzed. A description of the functions of the genes differentially expressed in at least 25% of the studies (six times or more), representing 41 genes, can be observed in Appendix A. The high number of differentially expressed genes concerning IMiD exposure demonstrated the widespread impact these drugs can exert in the human cell, altering several biological processes that may result in thalidomide embryopathy, when this exposure occurs during embryonic development. It is reasonable to hypothesize that this widespread effect on gene expression might result from the IMiD-induced TFs’ degradations, unbalancing the transcription regulatory processes.

Despite the focus of this study being the evaluation of TF proteins, non-coding RNA (ncRNA) data were not excluded from the transcriptome analysis. There were 154 long non-coding RNAs (lncRNAs), 12 small nucleolar RNAs (snoRNAs), five small nuclear RNAs (snRNAs), and two microRNAs (miRNAs) differentially expressed in at least two datasets. The lncRNAs, named *MALAT1* and *ST7-AS1*, were differentially expressed in five studies. *MALAT1* can act as a transcriptional regulator of genes involved in cell cycle and cell migration, whilst *ST7-AS1* does not have ontologies described, but has been associated with glioma.

### 2.4. Cell Cycle and Angiogenesis Are among the Top Biological Processes Altered in IMiD Exposure

To better understand the molecular roles of these differentially expressed genes, a gene-set enrichment analysis (GSEA) was conducted, ranking the genes by the number of times they were differentially expressed and evaluating the gene ontologies (GOs); the highest ranked genes were included. Regarding the biological processes, the mitogen-activated protein kinase 1 (MAPK) cascade was the more enriched ontology in the GSEA analysis (Appendix A); this ontology refers to an important pathway of intracellular signaling transduction. Other processes already known to be affected by IMiDs were also highly enriched, such as the cell cycle and angiogenesis ontologies, blood vessel development, and vasculature development (Figure 3). The cellular component mostly enriched was the nuclear chromosome, while the molecular function was chromatin binding (Appendix A). The GSEA analysis was consonant with the literature review, demonstrating that the main biological processes affected by IMiD exposure were the cell cycle- and angiogenesis-related mechanisms. In addition, the molecular function and cellular components enriched in the analysis helped to support the assumption of a systemic effect initiated by the degradation of TFs. This degradation might end up deregulating other biological processes that depend on strictly regulated transcriptions.

### 2.5. SP1, a C2H2 Transcription Factor, Is a Strong Candidate for an IMiD Neosubstrate

To evaluate the TFs that regulate differentially expressed genes, a TF-gene analysis was performed in the TRRUST database, which only comprised experimental data from humans. Since gene expression is very dynamic, genes differentially expressed in only one dataset, without previous literature reports of an IMiD-induced effect, were filtered out from the subsequent analyses. Hence, after these filters were applied, 3005 genes were queried in the TRRUST database, 2947 differentially expressed in two or more datasets, plus 58 genes differentially expressed once, but with a previous literature report regarding an IMiD-induced effect. As described previously, 119 of the genes provided in the literature were never differentially expressed in the 23 datasets analyzed, being also ruled out from the analysis (Appendix A). As a result, the TRRUST tool suggested 203 TFs experimentally known to regulate the expression of the 3005 queried genes. These 203 TFs were crossed with the list of proteins obtained from the literature review and with a list of C2H2 TFs, aiming to comprehend their function better; a Venn diagram presenting the intersection between the three lists is presented in Figure 4A. Two TFs suggested by TRRUST were C2H2 TFs already reported in the literature: SALL4 and SP1. The other four TFs suggested by TRRUST were non-C2H2 TFs; however, there are previous reports in the literature suggesting that they can be affected by IMiD exposure: CTNNB1, hypoxia-induced factor subunit alpha (HIF1A), nuclear factor kappa B subunit 1 (NFKB1), and Spi-1 proto-oncogene (SPI1). As previously described, CTNNB1 was considered an essential protein in the network analysis. The other five TFs were also presented in the network generated (Figure 2): SALL4 was related to the CRBN-CRL4 degradation property (yellow), while SPI1 and NFKB1 were related to cell-cycle mechanisms (blue), and HIF1A and SP1 were related to angiogenesis mechanisms (orange). Of the 203 TFs suggested by TRRUST, SP1 had the greatest number of targets, reported to regulate 127 genes from the list provided to TRRUST. SP1 targets included *VEGFA*, the gene of chemokine ligand 2 (*CCL2*), which was differentially expressed in seven of the 23 datasets analyzed, and cytochrome P450 family 1 subfamily B member 1 (*CYP1B1*), deregulated six times in the DGE analysis. A gene ontology over-representation analysis was performed with SP1 target genes, pointing to an involvement of these targets in proliferation/apoptosis processes and angiogenesis. SP1 itself was involved in angiogenesis, as well as *VEGFA*, *CYP1B1*, and 25 more of its target genes.

The TRRUST analysis pointed out SP1 as the TF that regulated the expression of the greatest number of genes encountered as being differentially expressed. SP1 is a C2H2 TF previously reported in the literature as being affected by IMiD exposure. Several of the SP1 target genes were related to angiogenesis and proliferation/apoptosis processes. Hence, it was reasonable to suggest SP1 as a probable neosubstrate of IMiD-induced degradation.

### 2.6. C2H2 Transcription Factors’ Roles in Embryonic Development Must Be Prioritized in the Search for IMiDs Neosubstrates

As presented in Figure 4A, 17 TFs suggested by TRRUST are C2H2 TFs never-before reported in the literature as being affected by the IMiDs. To verify whether they played a role in embryonic development, a gene ontology over-representation analysis was performed (Appendix A). It was observed that these TFs were overrepresented in the development of several embryonic structures and organs, especially the eye and heart, which are known to be part of the spectrum of congenital anomalies reported in thalidomide embryopathy (Figure 4B). The analysis demonstrated these TFs were also included in relevant pathways in embryonic development, such as Wnt- and Notch-signaling pathways. A summary of the main processes related to these 17 TFs analyzed is available in Table 3. Interestingly, many of these TFs belong to the same families, such as the Kruppel-like family (KLF), here represented by KLF2, KLF4, KLF6, and KLF8; the snail family transcriptional repressor (SNAI), with the members SNAI1 and SNAI2; zinc-finger E-box binding homeobox (ZEB), with the members ZEB1 and ZEB2; and the specificity protein (SP), with the members SP2 and SP3, the same family from SP1, reported previously. This analysis provided several TFs that might be neosubstrates of the IMiDs, including members of the same family as SP1. The IMiD-induced degradation of these TFs would lead to several alterations in gene expression, a strictly regulated process in embryonic development; this mechanism could be the key to explaining the occurrence of IMiD-induced congenital anomalies in organs, such as the heart and eyes.

### 2.7. Beta-Catenin and SP1 Might Be Essential to Explaining the Systemic Effects of IMiDs

As described above, the TRRUST analysis provided six TFs previously reported in the literature, the ones with a C2H2 domain, SALL4 and SP1, and the non-C2H2 CTNNB1, HIF1A, NFKB1, and SPI1, all of them presented in the analyzed protein–protein interaction network (Figure 2). Here, a second network was created, including the proteins provided in the literature and the 17 non-reported C2H2 TFs that were related to embryonic development gene ontologies (Figure 5). This new network was assembled similar to the previous one, queried in STRING, and transferred to Cytoscape for the topology analysis. Of the 17 TFs presently included in the study, eight presented interactions with the other nodes of the network were labeled in blue in the image. EGR1, KLF4, KLF6, SP3, and YY1 interacted directly with SP1 and CTCF indirectly, through YY1. SNAI1 and SNAI2 were included through their interactions with GSK3B. The colors of the nodes in this network emphasized the TFs. The C2H2 TFs are represented in green, while the non-C2H2 TFs are in pink; the gray nodes are the non-TF proteins. A topology analysis was also conducted via cytoHubba and used to represent the sizes of the nodes. According to this analysis, CTNNB1 continued to be the hub of the network and the most essential protein, presenting the highest-ranking values in degree, betweenness, closeness, and MCC criteria (Table 4). In addition to CRBN and CUL4A, in this analysis, SP1 presented high scores in the centralities criteria and could be considered an essential protein as well. This change occurred because six of the eight TFs added interacted with the whole network through SP1, directly or indirectly. The network configuration demonstrated that an IMiD-induced degradation of SP1 could negatively impact several C2H2 TFs that were active in embryonic development. Likewise, SP1 degradation could easily reflect on non-C2H2 TFs, such as HIF1A, NFKB1, and, especially CTNNB1, hence being spread to other proteins involved in diverse molecular mechanisms. Hence, these results suggest SP1 as a feasible IMiD neosubstrate and, once again, beta-catenin is an essential protein in the systemic role of IMiDs; however, it is not necessarily a neosubstrate for drug degradation.

## 3. Discussion

The present study, based on transcriptome and systems biology-allied strategies, aimed to prioritize TFs that could help to explain the effects of IMiDs on embryonic development. A systematized literature review was performed, annotating the proteins and genes altered/affected by the IMiDs. The annotated proteins were studied regarding their role as TFs through systems biology strategies, while the genes were evaluated together with 23 publicly available transcriptome datasets (Appendix A). The main conclusions were: (I) IMiDs have a widespread effect on gene expression that might be explained by their induced TF degradation; (II) beta-catenin (CTNNB1) is an essential node and hub of the network of IMiD-altered proteins; (III) SP1 is a putative neosubstrate of IMiD-induced TF degradation; and (IV) there are 17 C2H2 TFs known to regulate the expression of genes altered by IMiDs that are active in embryonic development and have not been studied previously.

A transcription factor is a definition applied to describe proteins involved in the regulation of the transcription process; hence, they are able to affect the expression levels of a gene [27]. Several processes, such as embryonic development and cell differentiation, are regulated by TFs [28,29]. IMiD-induced alterations in some TF proteins have been known for a while [17,30,31]; however, only recently, studies have suggested an IMiD-induced degradation of these TFs, driven by the binding of drugs to the CRL4-CRBN complex [19,20]; this mechanism is believed to be mostly centered, but not exclusive, to TFs with a C2H2 protein domain [21,25]. However, few TFs have been identified when studying the C2H2 degrome induced by IMiDs [23]. This is an alarming result; however, it is a potential overestimation of the IMiDs’ impact on gene expression. The evaluation of the several datasets included here can lead to an overestimation because of the high heterogeneity between the studies, inherent of the methodologies and study designs. Nevertheless, IMiDs’ potential effects on gene expression must be evaluated with great attention because gene expression regulation during embryonic development is known to be synchronized, complex, and deeply regulated [32]. TF degradation, not necessarily strict to C2H2 TFs, is a plausible explanation for the IMiDs’ impact on gene expression. Regarding non-protein-coding genes, IMiDs’ effects on ncRNAs have been little studied. According to the literature review, the genes altered by the IMiDs in this study were protein-coding genes (Appendix A). Although transcriptome studies should cover the whole amount of RNA in the cell, usually, they are designed to target mRNA [33]. In this regard, microarray probes were mainly designed to cover the protein-coding genes [34]; RNA-Seq library preparation was preceded by RNA prioritization steps, which included poly-A selection and rRNA depletion [35]. Even though not all researchers performed poly-A selection, the abundance of ncRNAs was small in relation to the mRNA; hence, the ncRNA was poorly captured in transcriptome studies [35]. Because of this limitation, targeted ncRNA transcriptome analyses were conducted when these molecules were the main objective of the research [36]. Although we did not exclude ncRNA studies, we did not encounter transcriptome datasets that specifically targeted these molecules or that were registered in the GEO database as “non-coding RNA profiling” experiments in IMiD exposure. Nevertheless, 173 ncRNA genes were differentially expressed at least twice, according to the transcriptome analysis performed here. Since the evaluation of ncRNAs was impaired due to their abundance compared to the mRNA, the results presented here might be a small representation of the overall impact of the IMiDs in non-protein-coding genes.

From a systems biology perspective, TFs exert a protein–gene interaction by binding to sequence-specific DNA motifs. TFs are also known to be part of distinct regulatory networks, mediating protein–protein interactions [27,37] and defining both target gene selectivity and chromatin dynamics [37]. In the present study, systems biology analyses suggested beta-catenin as an essential protein in the network of IMiD-altered proteins. Thalidomide is known to diminish beta-catenin expression [31] and a pharmacogenetics study associated variants in the beta-catenin gene, *CTNNB1*, with lenalidomide adverse effects in multiple myelomas [38]. Beta-catenin is part of the Wnt canonical pathway, involved in a series of embryonic development events, including the establishment of the body axis and the orchestration of tissue and organ development [39]. This regulation is based upon the integration of two distinct beta-catenin functions: structural, as a cell-adhesion junction molecule, and signaling, as a TF [40]. In addition, the abundance of beta-catenin-binding partners provides an interaction with other TFs and signaling pathways [39]. Hence, beta-catenin deregulation can alter several regulatory networks, helping to explain the IMiDs’ systemic effects (Figure 6). One protein that has a direct interaction with beta-catenin is SP1, a C2H2 TF. Thalidomide was shown to inhibit SP1 activity in endothelial cells [41], which led to two hypotheses of SP1 involvement in thalidomide embryopathy: SP1 can be a second messenger of growth factors involved in limb development [42], or SP1 transcriptional activity can be blocked by thalidomide, impeding this TF from binding to its motif [41]. Studies evaluating SP1 in light of the degrome evidence have not been encountered. Nevertheless, the findings in this study point to SP1 as a feasible neosubstrate of IMiD-induced TF degradation. First, there are 127 genes experimentally known to be regulated by SP1 that were differentially expressed in this study, including well-established genes, such as *VEGFA*; no other suggested TFs presented so many target genes with their expressions altered. Second, one of the main mechanisms involving altered genes is angiogenesis, a process that SP1 is known to be part of [41]. Third, SP1 interacts directly with other TFs known to be affected by IMiDs, including beta-catenin, NFKB1, and HIF1A, in addition to other suggested C2H2 TFs that can also be a part of the degrome.

According to Mackeh et al. (2018), 3% of human genes refer to C2H2 zinc-finger proteins, totalizing more than seven hundred proteins described [43]. In chicken embryos, a species well-established as an animal model of thalidomide embryopathy, C2H2 was suggested as one of the most dominant types of TFs in embryogenesis [44]. Despite the fact that the role of human C2H2 proteins is yet to be fully explained, most of the 17 C2H2 TFs suggested here were involved in embryonic development [45,46,47]. They are part of families known to have a role in this period, such as SNAI, ZEB, and KLF [45,46,47], in addition to other factors, such as GLI1 [48]. YY1 is also considered essential to embryogenesis [43]. For the completion of this study, SALL4 was the only C2H2 zinc-finger TF with a well-known role in limb development that was known to be degraded by the IMiDs [49]. SALL4 pathogenic variants lead to Duane-radial ray syndrome [50], an autosomal dominant syndrome that presents a pattern of limb anomalies very similar to the ones identified in thalidomide embryopathy [51,52]. SALL4 degradation might explain the typical limb anomalies caused by thalidomide exposure in utero; however, it is rational to affirm new C2H2 targets, named neosubstrates, and should be studied. From the perspective of an IMiD degrome, the TFs suggested here might help us to comprehend the effects of drugs in other organs and systems known to be affected by the embryopathies. Another interesting result was *CYP1B1* as a gene deregulated in six datasets analyzed. Despite being mainly involved in drug metabolism, CYP1B1 is known to promote angiogenesis by suppressing NF-kB activity [53]. In cancer cell studies, SP1 was considered a key mediator of CYP1B1 action, whilst CYP1B1 was shown to activate a epithelial to mesenchymal transition and Wnt/beta-catenin-signaling pathways, upregulating beta-catenin and other TFs, such as ZEB2 and SNAI1 [54]. These mechanisms must be studied in light of embryonic development to understand whether this regulation can also help to explain IMiDs’ effects on embryos. Nevertheless, it was established that embryonic development and oncogenesis presented several similarities, especially regarding the biological processes and molecular pathways involved [55].

One limitation of the present study was the lack of experimental validation of the prioritized TFs suggested. Nevertheless, all the analyses conducted in our study, from the literature review to the networks and TF–gene interactions, were only based on the experimental evidence; thus, no in silico prediction was performed. All the conclusions presented here were also based on the literature evidence regarding the biological processes that the genes encountered were known to be a part of. It is relevant to state that bioinformatics approaches were conducted well to highlight molecular mechanisms under different medical conditions and, therefore, should not be observed as pure theoretical, computational modeling [56,57]. Bioinformatics tools improve experimental validations by providing standardized pipelines to access cell and molecular mechanisms [56,57]. Molecular techniques using animal models and cellular methodologies can be very expensive and time-consuming [58]. Hence, the studies performed using public databases and providing a secondary analysis of omic data are helpful to better comprehend the biological processes altered after drug exposure, following the reduce, replace, and refine strategy in animal model research [59].

## 4. Materials and Methods

In summary, combined literature and database research was the basis for all the systems biology analyses, which included the evaluation of protein–protein interaction networks and protein–gene-targeted mechanisms. In protein–gene networks, the TFs were inserted as proteins and the gene its targets. Concomitantly, a second search was performed in the Gene Expression Omnibus (GEO) repository [60], annotating transcriptome studies that were evaluated regarding differential gene expressions (DGEs) in IMiD exposure. For a better comprehension, enrichment analyses and Venn diagrams were performed and created, respectively, throughout the study, using the data obtained from both DGE and systems biology strategies. Finally, the results were gathered through a network-assembling strategy, providing a protein–protein interaction network that presented the main TFs prioritized in the analyses. A detailed analysis description was presented in sequence. A scheme is presented in Figure 1.

### 4.1. Literature Review

A literature review was performed to annotate all the genes and/or proteins previously reported as disturbed by thalidomide, lenalidomide, or pomalidomide during embryonic development. The Rayyan tool [61] was used to annotate PubMed (Medline) and Embase manuscripts with the search terms available from Appendix A. The second search was performed in the PubTator repository [62], which was accessed through R language, using the *pubtatordb* package (https://github.com/cran/pubtatordb; URL accessed on 31 May 2023); Medical Subheading (MeSH) terms were applied to this research as follows: D013792 for thalidomide, D000077269 for lenalidomide, and C467566 for pomalidomide. The third literature search was performed using the Comparative Toxicogenomics Database (CTD) [63] by typing the name of the drug in the query search and downloading all the “References” provided.

Duplicates were excluded and the studies were primarily filtered by title and abstract screening. The selected manuscripts were fully read and the genes and/or proteins disrupted by the drugs we studied were annotated. For “disruption”, we considered any type of effect driven by the drugs (i.e., expression alteration, binding, activity inhibition, and interaction). Hence, the inclusion criteria for genes and/or proteins disrupted were: evidence of an experimental assay using human embryonic cells, tissues, or organoids exposed to one of the IMiDs, thalidomides, lenalidomides, or pomalidomides, and only statistically significant disrupted genes/proteins were included, according to the cutoffs used in the study. The exclusion criteria were: transformed cells, knockout studies, exposure to two or more concomitant drugs, and studies that had abstracts, but not the full manuscript, available.

### 4.2. Systems Biology Analysis

The proteins annotated in the literature review or the subsequent TF analyses were inserted in the STRING tool [64], comprising only experimental evidence of interactions and co-expressions with a score > 0.400. A network of protein–protein interactions was obtained and transferred to Cytoscape v.3.7.2 [65]. Topological network analyses, such as centrality measures, were calculated through the cytoHubba plugin [66]. Essential proteins were obtained through the maximal clique centrality (MCC) criteria. The MCC score was used to indicate the size of the node, i.e., MCC = 1 was represented by a node of size 10 and MCC = 5 was represented by a node of size 50. The colors of the nodes were selected manually, based on the network-relevant characteristics for each step of the analysis, and were detailed in the legends of figures.

### 4.3. Differential Gene Expression Analysis

Publicly available transcriptomes were downloaded from the GEO repository [60] and a DGE analysis was performed. Since gene expression was very dynamic and transcriptome analyses might result in several false-positive differentially expressed genes, a wide range of datasets was included. The terms “thalidomide”, “lenalidomide”, or “pomalidomide” were queried in GEO, filtered for *Homo sapiens*, and all the assays were annotated. The inclusion criteria were assays comprising human cells, tissues, or samples from patients who were exposed to one of the IMiDs (thalidomide, lenalidomide, or pomalidomide) at any age or stage of development. Datasets without raw data available, studies without a control group, knockout studies, and exposure assays or therapeutic schemes that used combined drugs in the same sample were excluded.

Microarray datasets were evaluated in R language by performing robust multi-averaging (RMA) normalization for Affymetrix studies with the *affy* package [67] and variance stabilizing normalization (VSN), followed by quantile normalization in Illumina or Agilent studies, using the NetworkAnalyst web interface [68] for sample processing. DGE was calculated through the *limma* package [69]. RNA-Seq samples were evaluated regarding quality control through FastQC software (https://www.bioinformatics.babraham.ac.uk/projects/fastqc/; URL accessed on 31 May 2023) [70]. Alignment and transcript count were obtained through the useGalaxy server [71], using Bowtie2 [72] and featurecounts [73], respectively. The alignment of all the samples was performed using the GRCh38 (hg38) reference genome. In R, the *edgeR* package [74] was used to perform a trimmed mean normalization (TMM) and calculate the DGE. For all the datasets, microarrays, and RNA-Seqs, a surrogate variable analysis (SVA) normalization was performed to remove batch effects. Genes with a logFC > |1| and an adjusted *p*-value < 0.05 were annotated for their significant differential expressions; *p*-value adjustment was performed by the false discovery rate (FDR). If none of the genes met the statistical criteria, the whole dataset was excluded from the present study.

### 4.4. Transcription Factor Analysis

The Transcriptional Regulatory Relationships Unraveled by Sentence-based Textmining (TRRUST) database [75] was used for the TFs analyses. TRRUST is a curated database for human TFs and their target genes. Aiming to reduce false-positive results, it was included in the TF search, only for genes that (I) were differentially expressed in at least two datasets, or (II) that were differentially expressed in only one dataset but had evidence of IMiD-induced alterations. Two distinct approaches for TF analyses were conducted. Proteins obtained in the literature review were inserted in the database and their target genes were annotated or, contrariwise, the set of the genes obtained by literature review or DGE strategies were inserted in the tool, then having their regulatory TFs annotated. In the second approach, only regulations with a *p*-value < 0.05 were considered significant. The complete list of human TFs was obtained from the study of Lambert et al. (2018) [27], available on the website http://humantfs.ccbr.utoronto.ca/ (accessed on 31 May 2023). The list of C2H2 TFs was downloaded from the HUGO Gene Nomenclature Committee (HGNC) [76]. Venn diagrams used in the analyses were performed in the tool provided by the Bioinformatics Evolutionary Genomics website, from Ghent University (http://bioinformatics.psb.ugent.be/webtools/Venn/, URL accessed on 31 May 2023).

### 4.5. Enrichment and Overrepresentation Analyses

Proteins and genes were obtained through a literature review, and TFs and their target genes were evaluated regarding gene ontologies (GOs), signaling pathways, and, specifically for TFs, protein domains. GOs and the Kyoto Encyclopedia of Genes and Genomes (KEGG) pathways were accessed through the *clusterprofileR* package [77], using the over-representation analysis strategy.

A gene set enrichment analysis (GSEA) was performed with the differentially expressed genes to rank the ontologies and pathways mainly affected by the IMiDs. Genes were ranked according to the number of studies where they were significantly differentially expressed. Hence, a gene differentially expressed in four studies received a score = 4, while a gene differentially expressed only once received a score = 1; consequently, genes that never presented differential expressions received a score = 0. Three types of ontologies were accessed: biological processes, molecular functions, and cellular components. According to the GO resource, molecular functions referred to the molecular activities performed by the gene products that, when evaluated together, resulted in a biological process. Cellular component referred to the anatomical location of the protein in the cell.

## 5. Conclusions

In addition to the previous fundamental studies that identified crucial molecular mechanisms behind thalidomide embryopathy [15,17,18], the systems biology approach presented here allowed us to evaluate these hypotheses in a more integrative manner. Moreover, this was also the first study to provide a systematized, strictly performed literature review on the genes and proteins altered by IMID exposure in embryonic development. Furthermore, 35 transcriptome datasets were processed and evaluated regarding differential gene expressions, making this research the most thorough one known to address IMiDs’ transcriptional effects. The transcriptomics strategy conducted here can also be applied to other drugs, aiming to evaluate the therapeutic or adverse effects, such as teratogenesis. In conclusion, not only did this study prioritize SP1 and beta-catenin as strong candidates for IMiD effects on embryonic development, it also analyzed the large amount of publicly available data, indicating there is much new knowledge to be integrated to understand an old challenge.

## Figures and Tables

**Figure 1 ijms-24-11515-f001:**
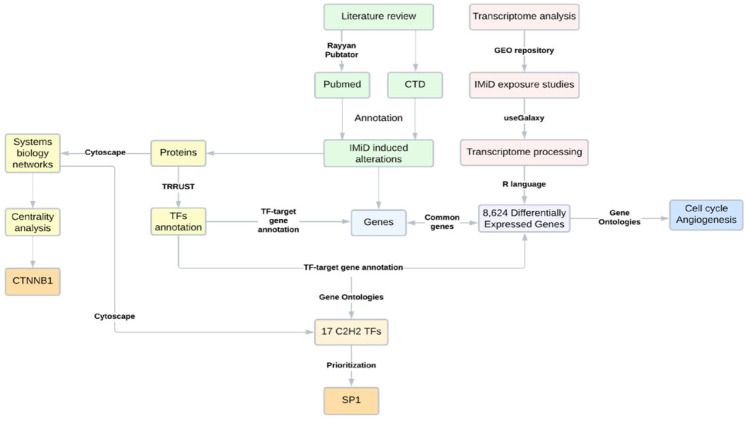
Scheme of the research strategy applied, comprising a literature review, bioinformatics, and systems biology analyses. CTD = comparative toxicogenomics database.

**Figure 2 ijms-24-11515-f002:**
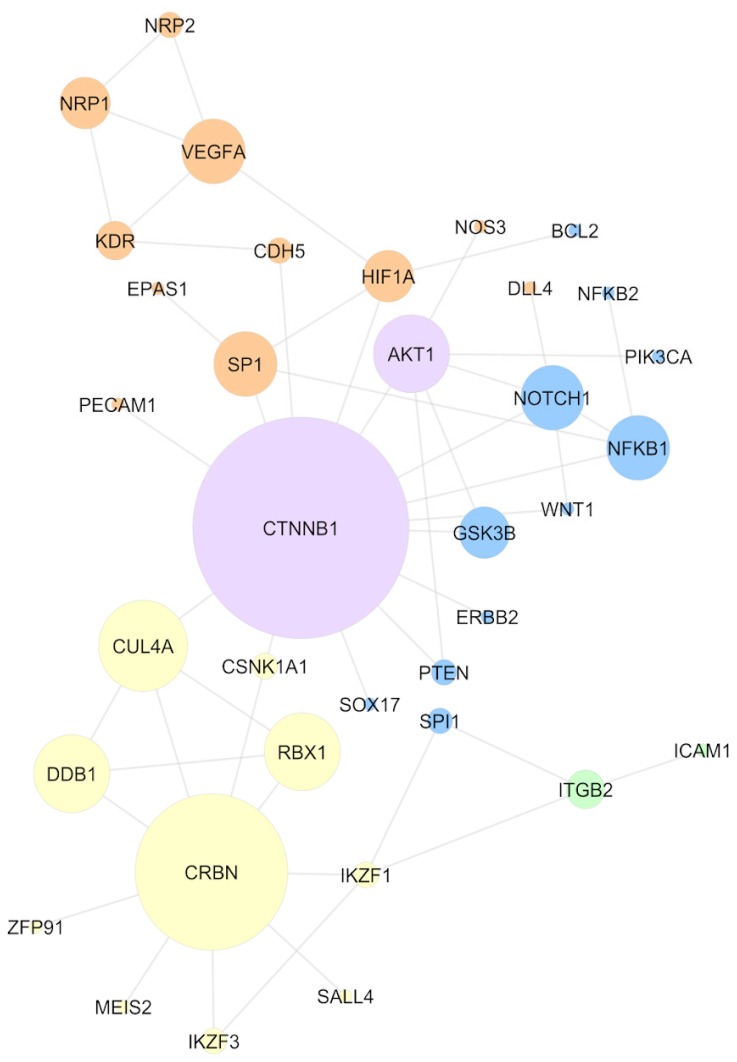
Network comprising experimental evidence of the protein–protein interaction, considering only IMiD-altered proteins, according to the literature review. Node size represents the maximal clique centrality (MCC) score. The colors of the nodes represent the mechanisms mainly described in the literature for the selected proteins: CRL4-CRBN binding (yellow), angiogenesis (orange), cell cycle (blue), and cell adhesion (green). AKT1 and CTNNB1 are presented in purple because of the association with multiple mechanisms.

**Figure 3 ijms-24-11515-f003:**
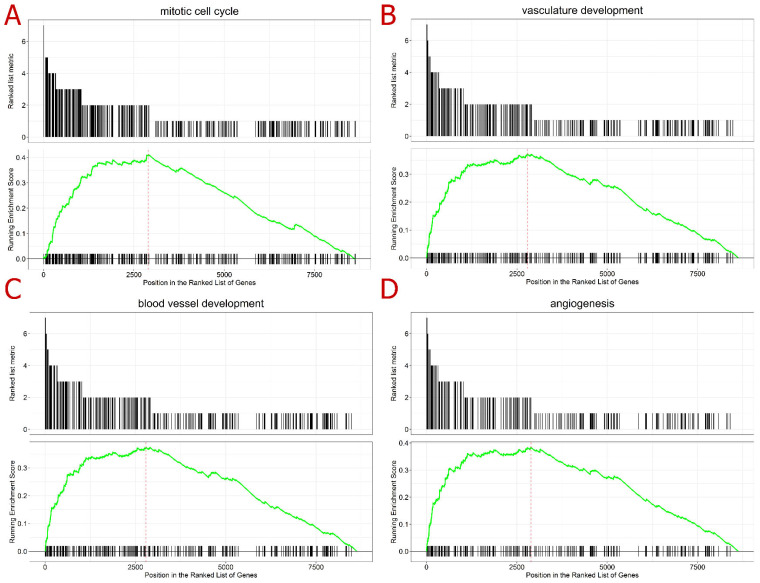
Gene set enrichment analysis (GSEA) top-ranked ontologies in the following order: (**A**) mitotic cell cycle (upper left), (**B**) vasculature development (upper right), (**C**) blood vessel development (lower left), and (**D**) angiogenesis (lower right). Legend: The first plot (ranked list metric) represents the number of datasets where the gene was differentially expressed, according to the 23 transcriptome datasets analyzed here; hence, the values are presented from 9 to 1. The second plot (running enrichment score) refers to the GSEA performed: the black lines along the x-axis represent the position of the gene in the ontology analyzed; the green line represents the running enrichment score for each of the genes; and the red line represents the maximum enrichment score for the analysis.

**Figure 4 ijms-24-11515-f004:**
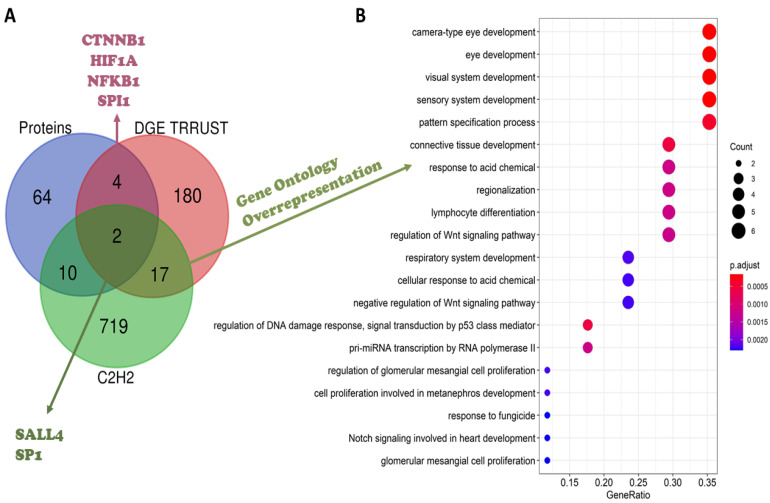
(**A**) Venn diagram considering the IMiD-affected proteins, according to the literature review (blue), transcription factors (TFs) provided by the TRRUST database (red), and the list of C2H2 transcription factors (green). (**B**) Over-representation analysis of the 17 C2H2 TFs suggested by the TRRUST analysis.

**Figure 5 ijms-24-11515-f005:**
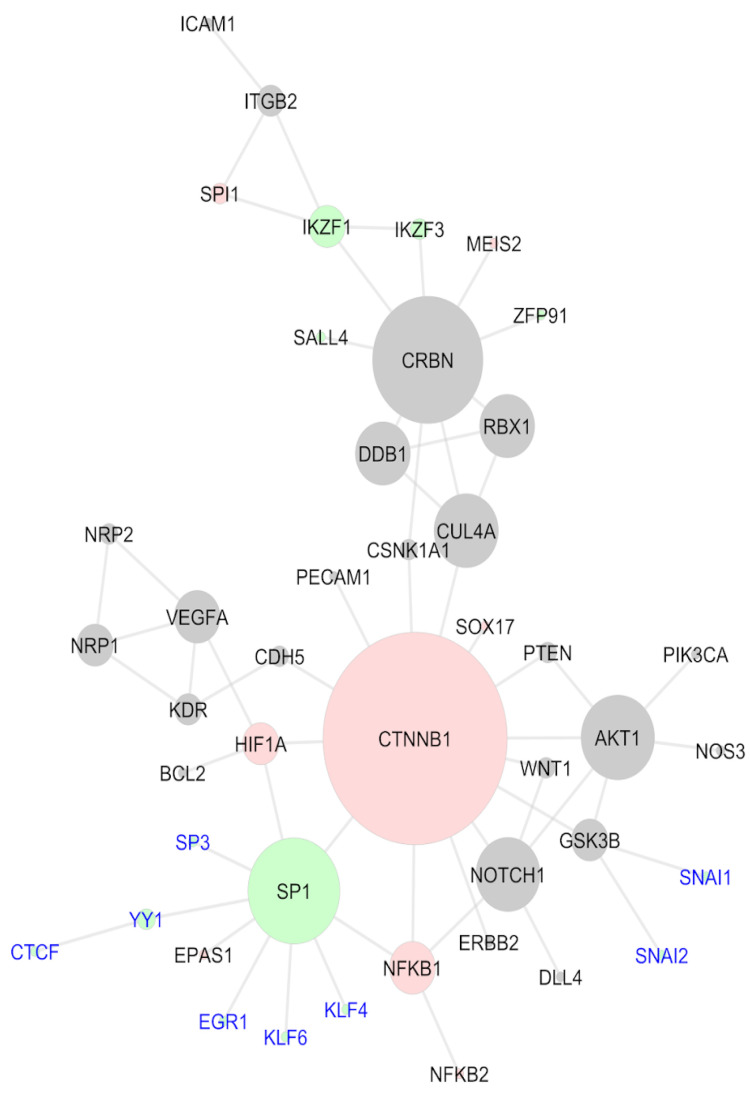
Systems biology network comprising the literature review proteins and the C2H2 transcription factors non-described as IMiD-affected that have a role in embryonic development; non-TF proteins (gray), non-C2H2 TFs (red), and C2H2 TFs (green). New TFs added are labeled in blue.

**Figure 6 ijms-24-11515-f006:**
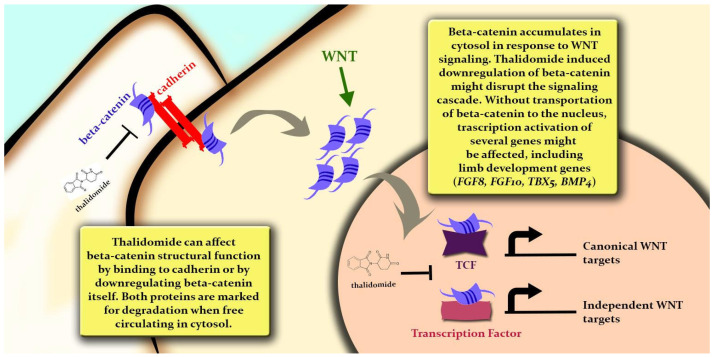
Representation of beta-catenin (CTNNB1) molecular mechanisms, with possible IMiD-altered effects (represented by thalidomide) and putative consequences on embryonic development.

**Table 1 ijms-24-11515-t001:** Topological analysis of the protein networks and centralities’ definitions.

Topological Analysis	Definition	Highest-Ranked Nodes (Literature Network) ^1^
Degree	Number of edges that connect to a node. Nodes with a high degree are defined as hubs.	CTNNB1 (13), CRBN (9), and AKT1 (5)
Closeness	Measures how fast the flow of information travels through the nodes. High closeness centrality scores indicate rapid information flow.	CTNNB1 (20.83), CUL4A (15.67), and CRBN (15.57)
Betweenness	Demonstrates how crowded a network is. High betweenness centrality score indicate nodes that can control the information flow.	CTNNB1 (713.67), CRBN (283), and CUL4A (220)
Maximal Clique Centrality (MCC)	Maximal clique indicates subsets of nodes that cannot be extended by adding additional nodes, because all the nodes in the mentioned subset are already interacting with each other. This centrality is proposed by the developers of cytoHubba as the best method to obtain the essential proteins in a network.	CTNNB1 (17), CRBN (12), and CUL4A (7)

^1^ Network available in Figure 2.

**Table 2 ijms-24-11515-t002:** Proportion of genes differentially expressed considering the reference genome and the number of datasets in which the genes presented differential expressions.

Number of Datasets with Differential Expressions	Number of Differentially Expressed Genes (%)
Total number of genes in the GRCh38 (human reference genome)	28,395 (100%)
1 or more	8624 (30.4%)
2 or more	2947 (10.2%)
3 or more	1041 (3.6%)
4 or more	334 (1.2%)
5 or more	118 (0.4%)
6 or more	41 (0.15%)
7 or more	16 (0.05%)
8 or more	3 (0.01%)
9	1 (0.003%)

**Table 3 ijms-24-11515-t003:** Groups and functions of the 17 C2H2 prioritized transcription factors (TFs).

TFs	Chromosomal Location	Gene Groups ^1^	Related Pathways or Ontologies ^2^
BCL6	3q27.3	BTB domain	Cytokine signaling in immune system
CTCF	16q22.1	Cilia/flagella	Nervous system development
EGR1	5q31.2		Regulation of cell survival, proliferation, and cell death
GLI1	12q13.3		Signaling by Hedgehog
HIC1	17p13.3	BTB domain	Metabolism of proteins
KLF2	19p13.11	KLF transcription factors	Embryonic and induced pluripotent stem cells and lineage-specific markers
KLF4	9q31.2	KLF transcription factors	FOXO-mediated transcription
KLF6	10p15.2	KLF transcription factors	Adipogenesis
KLF8	Xp11.21	KLF transcription factors	Epithelial to mesenchymal transition
SNAI1	20q13.13	SNAG transcriptional repressors	Gastrulation
SNAI2	8q11.21	SNAG transcriptional repressors	Epithelial to mesenchymal transition
SP2	17q21.32	Sp transcription factors	Histone deacetylase binding
SP3	2q31.1	Sp transcription factors	Metabolism of proteins
WT1	11p13		Nervous system development
YY1	14q32.2	INO80 complex	ESR-mediated signaling
ZEB1	10p11.22	ZF class homeoboxes and pseudogenes	Cytokine signaling in immune system
ZEB2	2q22.3	ZF class homeoboxes and pseudogenes	TGFB-receptor signaling

^1^ All the TFs described are also part of the zinc-finger C2H2 type group (HGNC, 2023). ^2^ Other than transcription regulation/gene expressions.

**Table 4 ijms-24-11515-t004:** Topological analysis of the protein networks and centralities’ definitions.

Topological Analysis	Definition	Highest-Ranked Nodes (Literature + New TFs) ^1^
Degree	Number of edges that connect to a node. Nodes with a high degree are defined as hubs.	CTNNB1 (14), CRBN (9), and SP1 (9)
Closeness	Measures how fast the flow of information travels through the nodes. High closeness centrality scores indicate rapid information flow.	CTNNB1 (26.37), SP1 (21.98), and CUL4A (19.67)
Betweenness	Demonstrates how crowded a network is. High betweenness centrality score indicate nodes that can control the information flow.	CTNNB1 (1325.67), CRBN (601), and SP1 (553.34)
Maximal Clique Centrality (MCC)	Maximal clique indicates subsets of nodes that cannot be extended by adding additional nodes, because all the nodes in the mentioned subset are already interacting with each other. This centrality is proposed by the developers of cytoHubba as the best method to obtain the essential proteins in a network.	CTNNB1 (20), CRBN (12), and SP1 (10)

^1^ Network available in Figure 5.

## Data Availability

The study only used data publicly available in databases and repositories. All the results generated are fully available in the Appendix A.

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
