# Peer review of "A New Strategy for the Old Challenge of Thalidomide: Systems Biology Prioritization of Potential Immunomodulatory Drug (IMiD)-Targeted Transcription Factors"

_ijms, 2023, doi:10.3390/ijms241411515_

Round 1
Author Response
Dear reviewer,
We are very grateful for all your contributions and have addressed all the issues presented, as comprehensively as possible.
Please see the attachment.

Reviewer 2 Report
This work is a metagenomic and bioinformatics analysis of the molecular mechanisms of Thalidomide Embryopathy from antiangiogenesis to oxidative stress to Cereblon-binding. The authors evaluate the transcription factors altered by immunomodulatory drugs (IMiDs), prioritizing the ones associated with embryogenesis through transcriptome and systems biology allied analyses.
The work as a whole is quite detailed and of sufficient value.
The text is well written and there are no special comments on the illustrations and the material assembled. Since there is quite a lot of data obtained, it is very difficult to study and verify it.
I see no limitations to the publication of this work, as all the primary data and approaches are presented and the conclusions obtained in the work are logical and justified.
Author Response
We appreciate and thank the reviewer for the comments and review. We performed alterations throughout the manuscript, as asked by other reviewers, but there were no alterations regarding the results obtained and conclusions.

Reviewer 3 Report
This manuscript explores systematically literature and molecular databases in search of transcription factors that are affected by the immunomodulatory drugs (IMiDs) thalidomide, lenalidomide and pomalidomide. The exploration is succesful in that it finds 17 C2H2 transcription factors not previousy described as sensitive to ImiDs, and demonstrate the central roles of CTNNB1 and SP1 in an interaction network that includes all these proteins. It has to be remarked, however, that all these findings are not approached experimentally, but they constitute valuable information as start point for additional investigation.
The manuscript is interesting, but there are issues to be considered.
MAJOR ISSUES
1.- Concerning the huge quantity of genes responsive to IMiDs found in this work (line 313), it seems that only protein-coding genes were found or were considered for the study. Isn’t there any evidence for the relevance of genes encoding non-coding RNA and their RNA products in relation with IMiDs?
2.- All the supplementary files: Figures S1 and S2, Supplemental Material 1, and each Supplemental Table S1-S8 should contain inside a full legend, including a title, a description of the content and how was obtained and/or used.
3.- Figure 2. The resolution of this image should be improved. The colors of low MCC score are difficult to see. Pink is difficult to distinguish from orange.
4.- Figure 3. The resolution of the image must be improved. Even more important, the meaning of the graphs is totally unclear. This figure should be explained in an intelligible manner.
5.- Figure 4B. The resolution of the image is very poor. It must be much improved. Like in Figure 2, the colors of low MCC score are very difficult to see.
6.- Throughout the manuscript there is an extensive use (abuse) of unnecessary saxon genitive. Below, in minor issues, the instances detected have been remarked. Most of them should be substituted by alternative expressions.
7.- In the references, the list of authors should be interrupted by “et al.” only when the number of authors is higher than a certain number, e.g. 6, 8, 10, but not just one author.
8.- A Conclusions section would be welcome.
MINOR ISSUES
Article title. Do not use just the IMiDs abbreviation. Substitute by “immunomodulatory drugs (IMiD)-targeted”. Actually, the IMiD abbreviation as short of “immunomodulatory drugs” is in my opinion confusing since it is also used for “Immune Mediated Inflammatory Diseases”. I think that IMD would be best as abbreviation of immunomodulatory drugs, however I understand that the confusing use of IMiD is much extended. This is the reason why I just recommend the minor change in the title.
Line 23. Change “degrade” to “induce degradation of”.
Line 29. Change “Protein’s” to “Protein”.
Line 29. CTNNB1 should be described in the abstract as beta-catenin, just as it is in line 76. Thereafter, the use of CTNNB1 or beta-catenin or both should be consistent. In the text, they are used interchangeably, what sometimes create certain confusion. In Figure 1, Table 1, Figure 2, Figure 4B and Figure S1, CTNNB1 should bear also the beta-catenine denomination.
Line 42. Change “sales around the world were” to “was” (thalidomide was used as a panacea, not the sales of thalidomide).
Line 43. Change “sooner” to “soon after”.
Line ro. Change “Thalidomide’s” to “Thalidomide”.
Line 59. Change “implicate” to “be involved”.
Line 67. The term “neosubstrates” is not common out of this particular field. Therefore, it should be defined at their first use or, perhaps, around line 62.
Legend to Figure 1. Define CTD database.
Lines 89, 113, 114, 158, 160, 167, 171, 183, 188, 209 and 240. Supplemental tables and figures should be cited as S1, S2, etc.
Line 102. Change “evidence of” to “evidence that”. Alternatively, you can say “evidence of” but in this case, in line 103, you should change “are more related” to “being more related”.
Line 105-106. This sentence is unclear. What “reports on proteins” are referred to? What do you mean “especially driven by research on CBRN? Also, please explain briefly what is the “degrome mechanism”.
Line 107. Change “IMiDs’ affected” to “IMiDs-affected”.
Line 130. Change “The nodes’ sizes” to “The size of the nodes”.
Line 131. Change “nodes’ colors” to “color of the nodes”.
Line 139. Change “can suggest” to “suggests”.
Lines 132-141. This includes only two long sentences difficult to read. Please split to shorter ones.
Line 141. Change “IMiDs’” to “IMiDs”.
Line 146. Change “The nodes’ colors” to “The color of the nodes”.
Line 153. Change “with being VEGFA” to “with VEGFA being”
Line 157. Change “of the samples’ low quality” to “the low quality of the samples”. And explain what is meant by “low quality”.
Line 157. Define DGE.
Line 259. Change “IMiDs’ systemic effects” to “systemic effects of IMiDs”.
Line 267. Insert a comma after included.
Line 270. Change “nodes’ color” to “color of the nodes”.
Line 273. Change “the nodes’ sizes” to “the size of the nodes”.
Line 274. Change “network’s” sizes” to “network”.
Line 285. Change “of these drugs’ degradation” to “for drug degradation”.
Line 294. Change “IMiDs’ effects” to “effects of IMiDs”.
Line 309. Change “drugs’ binding” to “binding of drugs”.
Line 312. Change “IMiDs’ C2H2 degrome” to “C2H2 degrome induced by IMiDs”.
Line 353-357. Long sentence. Please split it.
Line 358. Change “has been” to “is”.
Line 365. Change “drugs’ effects” to “effects of drugs”.
Line 427. Change “drug’s name” to “name of the drug”.
Line 445. Change “nodes’ size” to “size of the nodes”.
Line 446. Change “Nodes’ colors” to “The color of the nodes”.
Line 448. Change “figures’ legends” to “legends to figures”.
Line 449. It is stated that “gene nodes were represented as triangles”, but actually I do not find gene nodes represented by triangles anywhere in the manuscript of supplementary material.
Line 480-481. The criterium for inclusion (II) is not understandable: there is something missing between “dataset” and “added” in line 481.
Line 522. G.G.C. should be G.C.G.
Line 556. The names of the authors of reference 2 should not be fully higher case.
Moderate editing of English is required
Author Response

(The authors gave the same response as above.)

Reviewer 4 Report
Comments
The authors conducted a bioinformatic analysis of available RNA sequencing and protein expression data, identifying CYNNB1 and SP1 as potential degradation neo-substrates of IMiDs. This study offers valuable insights and references for individuals interested in understanding the molecular mechanisms underlying Thalidomide Embryopathy (TE). With the necessary revisions, the manuscript holds promise for publication.
1. It is odd to use “graph” as one of the keywords.
2. Please specify the number of overlapping genes in line 94.
3. In line 158-159, within these 8624 genes, how many of them are present in 2 or more datasets?
4. In line 238, please change “role” to “roles”.
5. In line 312, the conclusion appears to be potentially misleading and requires revision. Firstly, it should be noted that the 8,624 genes under consideration originate from various Differential Gene Expression (DGE) studies, each employing different experimental protocols and testing systems. Therefore, the lack of consistency in these factors may introduce variability and uncertainty into the results. Secondly, this study did not investigate the extent of differential expression exhibited by these genes. Consequently, the conclusion should be adjusted to reflect this limitation. Lastly, it is crucial to consider only the genes that overlap across multiple studies as true positive calls, as they provide more reliable evidence. Consequently, the reported 30.4% should be recognized as a potentially significant overestimation.
Author Response

(The authors gave the same response as above.)

Round 2
Reviewer 1 Report
1. All figures are still in very insignificant resolution.
2. All plots of Figure 3 should be arranged in one figure with all four different plots. For example, A) mitotic cell cycle B) vasculature development C) blood vessel development D) angiogenesis.
3. Figure 1 is under section 2.1; however, it should be moved above the section.
Author Response
We thank the reviewer for the comments and suggestions. We have addressed each comment as comprehensively as possible, as follows:
- All figures are still in very insignificant resolution.
We thank the reviewer for the observation. We believe the pdf generation is reducing the quality of the figures, because we submitted all the images in a 600dpi resolution, as asked by MDPI editorial team. We have also uploaded the figures in high quality in a Google Drive folder, so the reviewer can check the 600dpi version. Please access through this link: https://drive.google.com/drive/folders/1diyt0wksEJSQbbnYPqgbcfpsBz5GlyMR?usp=sharing
- All plots of Figure 3 should be arranged in one figure with all four different plots. For example, A) mitotic cell cycle B) vasculature development C) blood vessel development D) angiogenesis.
We thank the reviewer for the suggestion. The plots were reorganized, as asked by the reviewer.
- Figure 1 is under section 2.1; however, it should be moved above the section.
We thank the reviewer for the observation. Figure 1 was moved, as asked.
Reviewer 3 Report
All the changes requested by this reviewer have been satisfactorily completed, and the manuscript is considerably improved.
Minor editing of English language is required
Author Response
We thank the reviewer for the appreciation. An English revision was performed in the 2nd version of the manuscript, as asked.
Round 3
Reviewer 1 Report
Authors have improved the manuscript and addressed all the reviewer's comments.